# Influencing Towards Stable Multi-Agent Interactions

**Woodrow Z. Wang, Andy Shih, Annie Xie, Dorsa Sadigh**
Department of Computer Science, Stanford University
{wwang153, andyshih, anniexie, dorsa}@stanford.edu

**Abstract:** Learning in multi-agent environments is difficult due to the non-stationarity introduced by an opponent's or partner's changing behaviors. Instead of reactively adapting to the *other agent's* (opponent or partner) behavior, we propose an algorithm to proactively influence the other agent's strategy to *stabilize* – which can restrain the non-stationarity caused by the other agent. We learn a low-dimensional latent representation of the other agent's strategy and the dynamics of how the latent strategy evolves with respect to our robot's behavior. With this learned dynamics model, we can define an unsupervised stability reward to train our robot to deliberately influence the other agent to stabilize towards a single strategy. We demonstrate the effectiveness of stabilizing in improving efficiency of maximizing the task reward in a variety of simulated environments, including autonomous driving, emergent communication, and robotic manipulation. We show qualitative results on our website.

**Keywords:** multi-agent interactions, human-robot interaction, non-stationarity

## 1 Introduction

Reinforcement learning has shown impressive performance in recent years [1, 2], but much of the focus has been on single-agent environments. Many real-world scenarios, on the other hand, involve coordinating with multiple agents whose behavior may change over time, making the task non-stationary. For example, when playing volleyball, our teammate may play more aggressively over time if they are hitting the ball well, so we may need to step aside and set the ball for them more often. Many techniques in multi-agent reinforcement learning (MARL) have been proposed for dealing with such non-stationarity in the other agent's strategies [3].

One set of approaches reactively adapts to the other agent's change in strategy, for example by simply forgetting past experiences and training on more recent data [4, 5, 6, 7, 8]. Other works try to learn a model of the other agent and take into account how the robot's actions affect the other agent's strategies over time [9, 10, 11, 12, 13]. With knowledge of the other agent's strategy dynamics, we can take actions to influence the other agent in a way that maximizes our long-term reward [10, 13].

Although influencing other agents to maximize reward is a useful approach in principle, it can be challenging in practice. Indeed, if we have access to a faithful model of the other agent's strategy dynamics, then we should choose actions that carefully nudge their high-level strategy to maximize our long-term reward. However, learning and leveraging such a model of the other agent's strategy dynamics can be impractical. For one, even just estimating the other agent's current strategy is already difficult with limited data, let alone learning the dynamics between strategies. On top of that, the robot may have to figure out a best response for a large number of opponent strategies, making the full learning process extremely sample inefficient.

Instead, we should take into account the difficulty of learning for our ego agent. [1] That is, we would still like to influence the other agent in a useful way, but in a manner that minimizes the burden of the ego agent's learning algorithm. *Our key insight is that we can stabilize the other agent's strategy in a multi-agent environment to reduce non-stationarity and facilitate learning a best response to the other agent.* Our approach is a specific form of influencing – learning how to stabilize – that

---

[1]We refer to the other agent interchangeably as the opponent, regardless of whether they are competitive or cooperative [3]. Also, we interchangeably refer to our robot as the ego agent.

5th Conference on Robot Learning (CoRL 2021), London, UK.

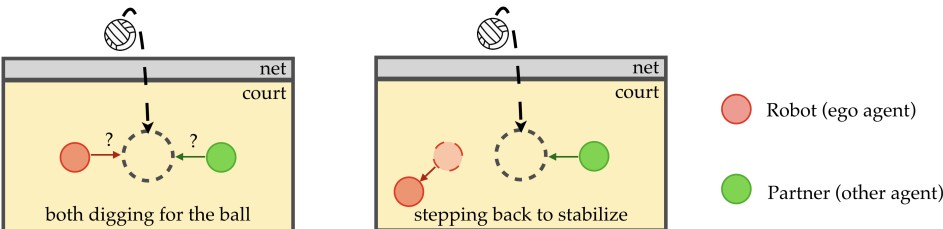

Figure 1: Coordination with a partner can be critical to succeeding in a task. In this volleyball game, the red and green agents both can either dig for the ball or step back for their partner. (Left) A partner that indecisively switches between digging and staying back can be extremely difficult to coordinate with, potentially costing the match. (Right) By deliberately stepping back, the red agent can stabilize the partner's strategy to dig for the ball, developing a convention that allows for easier learning of how to win the volleyball match.

aims to reduce the non-stationarity of the other agent, and in turn makes learning easier for the ego agent. Intuitively, stabilizing the other agent's strategy allows us to avoid learning about complex high-level strategy dynamics, and focus our efforts on maximizing reward against a fixed strategy.

As an example of stabilizing the other agent, imagine again that we are playing beach volleyball with a teammate. When the ball comes perfectly in between our teammate and us, it is not obvious which of us should dig for the ball (see Fig. 1 (left)). In principle, it is possible to carefully model the dynamics of how our teammate's strategy changes based on our actions and precisely predict who should get the next ball that falls in between us. However, it can take many interactions before we learn a good model of our opponents, and constantly switching roles between digging for the ball and staying back can make learning difficult. Instead, we can stabilize our teammate by making our intentions clear: we actively take a step away from the ball so our teammate knows to dig for the ball, as in Fig. 1 (right). This simplifies the learning process by reducing the need to learn complicated strategy dynamics, and making the task more stationary.

In particular, our method combines the robot's task reward with a stabilizing reward, where stability is measured with respect to the other agent's strategies learned in an unsupervised way. The task reward pushes the robot to do well in the environment, while the stabilizing reward aims to make the learning process easy and sample efficient. We demonstrate the strength of our approach in 7 different tasks, including autonomous driving, an emergent communication task, and simulated robot environments.

Our main contributions in this paper are as follows:

- We propose augmenting our robot's reward function with a stability reward in order to explicitly encourage the robot to influence the other agent towards a stable strategy.

- We demonstrate that the stability reward can be defined as an unsupervised reward using learned representations of the other agent's strategy.

- We perform experiments in simulated environments demonstrating that stabilizing the other agent's strategy improves our robot's ability to coordinate and maximize the task reward.

## 2  Related Work

**Multi-Agent Reinforcement Learning.** Learning in multi-agent environments can be extremely difficult when the other agents have non-stationary behavior [14, 15, 16]. Hernandez-Leal et al. [3] provide a taxonomy of existing methods that attempt to deal with this non-stationarity issue in multi-agent environments. Some works seek to forget about previous experiences with old opponent strategies and improve adaptation by only training on more recent data [4, 5, 6, 7, 8, 17]. Further, centralized training has shown promise in multi-agent environments [18, 19]. Some works design explicit communication channels in which agents can share their policy parameters or gradient updates [20, 21]. However, these methods require that there exists some centralized method in which agents can communicate or share parameters. We are mainly interested in the decentralized setting where we train an ego agent to interact with other agents that we do not have control over.

**Opponent Modeling.** Instead of ignoring the opponent's changing strategies, prior opponent modeling works design agents that estimate their opponent's policies or parameter updates [22, 9, 23, 11, 12, 24, 25] in a potentially recursive manner. Peer-rewarding has also emerged as a method of encouraging cooperative behavior [26, 27, 16], but requires the assumption that an agent's reward function can be externally modified. Several works recognize that the ego agent's policy can directly induce change in the opponent's policies [13, 28, 29, 30]. In [10], the ego agent learns not only a representation of the opponents' strategies, but also learns a dynamics model for how the opponent's strategies change conditioned on the ego agent's behavior.

**Representation Learning of Latent Strategies.** Representation learning is critical to the tractability of opponent modeling. It may be computationally inefficient to learn all opponent intentions or explicitly learn the parameters of the opponent's policies. Instead, we recognize that there is often some underlying structure in the opponent policy space, which can be captured by a low-dimensional latent strategy. Prior works learn latent context variables to help distinguish between separate tasks, which could correspond to different opponents [31, 32, 33, 34, 35]. Representation learning has been shown to be useful in developing a dynamics model of the opponent's policy in multi-agent environments [36, 10]. In our work, we show that it can be beneficial to cluster latent strategy representations into a discrete latent space [37, 38] to identify shared structure among similar strategies.

**Stability and Surprise in Reinforcement Learning.** To mitigate the difficulty of learning in a non-stationary environment, we seek to stabilize the opponent's strategy – or minimize the surprise in the opponent's strategy. Recent works in reinforcement learning seek to do the opposite of designing maximum entropy methods to improve exploration [39, 40, 41]. Stability in the opponent's strategy can reduce the non-stationarity in the environment and improve efficiency of learning a best response [42]. Surprise minimizing reinforcement learning (SMiRL) trains an agent to seek predictable and repeatable states to counteract the environment's entropy [43]. SMiRL considers a long horizon notion of stability in which the state should be of high likelihood with respect to the distribution of states encountered so far. Our work is similar in principle to SMiRL; however, we consider a pairwise notion of stability between consecutive latent strategies, as we recognize that an opponent that unpredictably switches between a few strategies is highly non-stationary, yet each individual strategy can be high likelihood with respect to the overall distribution of previously seen strategies.

## 3 Problem Statement

We consider two-player multi-agent settings, where there is an ego agent that we have control over and another agent that we do not have direct control over. Since we only control the ego agent, we can consider our problem formulation in terms of a single-agent Markov decision process where the opponent policy is absorbed as part of the environment transitions and rewards. Non-stationarity in the opponent policy then corresponds to non-stationarity in the environment transitions and rewards.

**Hidden Strategies.** The agents repeatedly interact with each other across multiple interactions, with each interaction lasting multiple timesteps. We assume that the opponent policy $z$ is fixed within each interaction but can change across interactions. We consider environments with no hidden states, so that if the ego agent can observe the exact opponent strategy, then we can formulate the ego agent's task as a fully observable MDP. However, since the ego agent does not directly observe the opponent's strategy, we consider a Hidden Parameter MDP (HiP-MDP).

**Hidden Parameter Markov Decision Process.** We formulate the interactions as a variant of a HiP-MDP, which is a tuple $\mathcal{M} = \langle \mathcal{S}, \mathcal{A}, \mathcal{Z}, \mathcal{T}, \mathcal{T}^z, \mathcal{R}, H, \gamma \rangle$ [44], which includes an additional $\mathcal{T}^z$ that governs the dynamics of the opponent's strategies. We let $z^j \in \mathcal{Z}$ be the opponent strategy for the $j$-th interaction. At each timestep $t$, the ego agent is in state $s_t \in \mathcal{S}$. The ego agent takes action $a_t \in \mathcal{A}$ and the environment transitions to $s_{t+1}$ with probability $\mathcal{T}(s_{t+1}|s_t, a_t, z^j)$. The ego agent receives the task reward $\mathcal{R}_{\text{task}}(s_t, a_t, z^j)$. The interaction ends after $H$ timesteps and $\gamma$ is the discount factor. Both the transition and reward function are unknown. During interaction $j$, the ego agent observes the trajectory $\tau^j = \{(s_1, a_1, R_{\text{task}}(s_1, a_1, z^j)), \ldots (s_H, a_H, R_{\text{task}}(s_H, a_H, z^j))\}$. Notice that the ego agent does not observe the opponent's strategy $z^j$ in the trajectory, which makes learning the value of an observation and action difficult – the reward of an observation and action can change significantly as the opponent's strategy changes across interactions.

We consider the setting where the opponent's change in strategy is governed by the trajectory of the previous interaction. At the end of interaction $j$, the opponent's strategy transitions to $z^{j+1}$ based

on strategy dynamics $\mathcal{T}^z(z^{j+1}|z^j, \tau^j)$. Namely, the ego agent's behavior during interaction $j$ can directly influence the opponent's strategy for interaction $j+1$. Next, we define a notion of stability between interactions $j$ and $j+1$ as follows:

**Definition 1** *Let $z^j, z^{j+1} \in \mathcal{Z}$ be two opponent strategies. The strategies $z^j, z^{j+1}$ are **pairwise $\epsilon$-stable** with respect to a metric space $(\mathcal{Z}, d)$ if and only if $d(z^{j+1}, z^j) < \epsilon$.*

**Stabilizing Opponent Strategies by Influencing Across Repeated Interactions.** Over repeated interactions, the ego agent explores HiP-MDP $\mathcal{M}$ over a sequence of opponent strategies $(z^1, z^2, \dots)$. The ego agent's objective is to maximize the cumulative discounted sum of rewards over all interactions, shown in Eq. 1. The ego agent's ability to maximize the reward within each interaction is heavily dependent on their ability to estimate the opponent strategy $z^j$ for that interaction. Without knowing the opponent's strategy, the ego agent will have significant difficulty taking an action that is a best response to the opponent's actions. It can be prohibitive to learn a model of all the opponent's strategies and form a general policy for the ego agent that can react to any opponent strategy; however, if the ego agent can stabilize the opponent strategy to a fixed point, i.e., $\exists z^k$ such that $\mathcal{T}^z(z^{j+1}|z^j, \tau^j) > 0$ if and only if $d(z^j, z^{j+1}) < \epsilon, \forall j \in \{k \dots N\}$ (where $N$ is the total number of interactions), then the ego agent can more easily learn a best response to the opponent strategy in a stationary environment.

With $z$ fixed across interactions, the HiP-MDP is reduced to a fully observable MDP parameterized by $z$. At a high level, we are encouraging the agent to learn how to explore the HiP-MDP in a subset of the state space, where the hidden opponent strategy is kept stable between interactions. Thus, the ego agent is effectively learning in a fully observable MDP, given that the fixed – but unknown – opponent strategy $z$ is not changing. In terms of the exploration-exploitation tradeoff, stabilizing the opponent strategy would reduce the ego agent's required exploration in order to find a best response to the opponent's fixed strategy.

We note that it is not always easy or possible to *stabilize* the opponent's strategy. Stabilizing requires that the dynamics of the opponent's strategy are defined such that the ego agent can form an interaction $\tau^j$ where $\mathcal{T}^z(z^{j+1}|z^j, \tau^j) > 0$ if and only if $z^j$ and $z^{j+1}$ are pairwise $\epsilon$-stable. In general, as long as there is a fixed point $z^k$ where the optimal value function satisfies $V^*(s_1, z^k) \geq V^*(s_1, z)$ for any other $z \in \mathcal{Z}$, then stabilizing should not hinder the task reward in the long run.

# 4  Stable Influencing of Latent Intent (SILI)

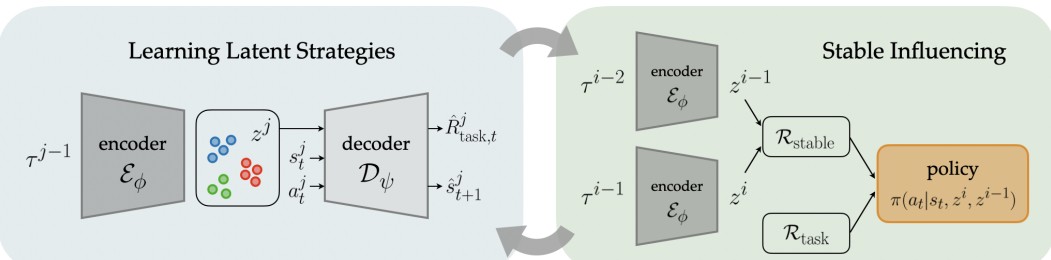

Figure 2: Stable Influencing of Latent Intent (SILI). (Left) We learn a dynamics model of the opponent's strategies conditioned on the ego agent's past trajectory. The latent strategy is learned in an unsupervised way by jointly learning a decoder to predict the state transitions and task rewards. (Right) We combine a stability reward with the task reward to train our policy using RL. The stability reward is unsupervised and defined by minimizing the pairwise distance between the previous two predicted latent strategies.

In this section, we present our approach of influencing the latent strategy of our opponent towards a stable strategy. At a high level, the ego agent learns an encoder that predicts the opponent's next latent strategy given the previous trajectory. The ego agent then conditions its policy for the next interaction on the predicted latent strategies of the opponent. But, since we never observe the ground truth latent strategies, we must learn the latent strategy representations in an unsupervised way. To do so, we follow the setup in [10], where we jointly learn a decoder that reconstructs the task rewards and transitions for the next interaction using the predicted latent strategy. Using this predicted latent

strategy, we can measure the stability of the opponent's latent strategy, and optimize for a weighted combination of the task reward and a stability reward.

**Learned Latent Strategies.** First, we describe the representation learning component of our method shown in Fig. 2 (left). Recall that we assume that the latent strategy dynamics follow $\mathcal{T}^z(z^{j+1}|z^j, \tau^j)$. We can approximate these latent strategy dynamics by learning an encoder $\mathcal{E}_\phi$ that maps trajectories $\tau^{j-1}$ to a latent representation $z^j$ of dimension $k$ (chosen as a hyperparameter). Here, we model the encoder as $f(z^j|\tau^{j-1})$ instead of $f(z^j|z^{j-1}, \tau^{j-1})$, as in [10]. This simplification, which often works well in practice, assumes that the previous trajectory $\tau_j$ already has enough information to reconstruct $z^{j+1}$.

Although we do not observe the ground truth latent strategies, we can still learn them in an unsupervised way, since we know that they should be predictive of the task reward and transitions. Thus, we jointly train a decoder that predicts the task reward and transitions from the encoded latent representation. In particular, for interaction $j$ and timestep $t$, the decoder $\mathcal{D}_\psi$ takes as input states and actions $s_t^j, a_t^j$ as well as the latent strategy embedding $z^j$, and reconstructs the next state $s_{t+1}^j$ and task reward $r_t^j$. To train the encoder and decoder, we use the following maximum-likelihood objective: $\max_{\phi,\psi} \sum_{j=2}^N \sum_{t=1}^H \log p_{\phi,\psi}(s_{t+1}^j, r_t^j | s_t^j, a_t^j, \tau^{j-1})$.

**Reinforcement Learning with Stable Latent Strategies.** As the opponent's strategy changes, the ego agent's policy should change in response. Naturally, we should condition the ego agent's policy on the opponent's latent strategies. When considering the goal of keeping the opponent's strategy pairwise stable, it is important that the ego agent conditions their policy on at least the most recent two predicted latent strategies since the stability reward depends on the distance between the consecutive latent strategies, as shown in Fig. 2 (right). Concretely, the ego agent's policy is defined as $\pi_\theta(a_t|s_t, z^j, z^{j-1})$, where $z^j = \mathcal{E}_\phi(\tau^{j-1})$.

We have defined the environment reward to be $\mathcal{R}_{\text{task}}$. We then define the stability reward to be the distance between consecutive predicted opponent strategies, where $d$ can be any generic distance metric over the latent space $\mathcal{Z}$ as follows: $\mathcal{R}_{\text{stable}} = -d(z^j, z^{j-1})$.

Here, we abuse notation and use $z^j$ to refer to the prediction of the opponent's strategy since we never observe the ground truth. Finally, our new reward function is defined as a weighted average of these objectives, parameterized by stability weight $\beta$: $\mathcal{R}_{\text{total}} = (1-\beta) \cdot \mathcal{R}_{\text{task}} + \beta \cdot \mathcal{R}_{\text{stable}}$. With $\beta = 1$, the ego agent's entire objective is to stabilize the opponent strategy, and with $\beta = 0$, the ego agent greedily maximizes the environment reward.

**Optimizing Rewards Across Interactions.** Our ego agent seeks to influence the opponent towards desirable latent strategies. With the encoder $\mathcal{E}_\phi$, the ego agent can deliberately take actions in trajectory $\tau^j$ in order to influence the opponent to a stable $z^{j+1}$.

To learn how to influence across repeated interactions, we train our ego agent's policy $\pi_\theta$ to maximize rewards across the repeated interactions

$$\max_\theta \sum_{j=1}^\infty \gamma^j \mathbb{E}_{z^j, z^{j-1}, \tau^j \sim \rho_{\pi_\theta}^j} \left[ \sum_{t=1}^H \mathcal{R}_{\text{total}}(s, a, z^j, z^{j-1}) \right], \tag{1}$$

where $\rho_{\pi_\theta}^j$ is the trajectory distribution induced by $\pi_\theta$ under the transition function $\mathcal{T}(s_{t+1}|s_t, a_t, z^j)$. With the total reward including the stability reward, the ego agent learns to take interactions such that the opponent follows a stable latent strategy so the ego agent can easily learn a best response to in turn maximize the task reward.

**Modeling Latent Strategies.** We examine three methods to model and measure distances between the changing latent strategies: continuous latent variables, discrete latent variables, and partial observations of the opponent's strategy. The ego agent never directly observes the opponent's strategy.

*1) Continuous Latent Variables.* In the most general case, the latent strategy space is allowed to be continuous. This enables the latent strategy space to represent an infinite number of possible opponent strategies. With the continuous latent variables $\mathcal{Z}$, we use the Euclidean norm for the stability reward: $\mathcal{R}_{\text{stable}} = -d(z^j, z^{j-1}) = -||z^j - z^{j-1}||_2$

*2) Discrete Latent Variables.* In this setting, we constrain the latent strategy space to be discrete categorical variables by using the Gumbel-Softmax distribution [37, 38]. Intuitively, we cluster the

continuous latent variable representations to filter the noise of measuring distance in the latent space. Further details of the discretization method can be found in Appendix A. With these categorical latent variables $\mathcal{Z}$, we use the discrete distance metric for the stability reward:

$$\mathcal{R}_{\text{stable}} = -d(z^j, z^{j-1}) = \begin{cases} -1 & z^j \neq z^{j-1} \\ 0 & z^j = z^{j-1} \end{cases} \tag{2}$$

We found empirically that discrete latent variables improve the consistency of the stability reward signal compared to continuous representations, despite the decrease in expressivity and granularity provided by the distance metric. Intuitively, it is easier to identify whether or not a strategy is similar enough to be clustered with another strategy than it is to exactly identify the distance between predicted strategies in a latent space.

*3) Partial Observations of Changing Strategy.* In this setting, we assume that the ego agent is able to observe if their opponent's strategy changes between timesteps, so no representation learning is necessary to stabilize. We shed the assumption that the latent strategies are fixed within an interaction, so we denote the latent strategy at a timestep $t$ as $z_t$. Thus, the stabilizing reward can be defined as $\mathcal{R}_{\text{stable}} = c_t = -\mathbb{1}[z_t \neq z_{t-1}]$, which indicates whether the opponent strategy has changed. This corresponds to settings where it may be easy to estimate when an opponent's strategy has changed, but it may be difficult to estimate their exact strategy [12].

## 5 Experiments

We perform experiments in a range of environments across different latent strategy space settings: continuous latent variables, discrete latent variables, and partial strategy observations. We compare our approach, Stable Influencing of Latent Intent (SILI) to various multi-agent learning baselines with implementation details described in Appendix A:

**Oracle.** Oracle is given the true opponent's strategy as an observation.

**Learning and Influencing Latent Intent (LILI) [10].** LILI combines soft actor-critic (SAC) [45] with a representation learning component to model the opponent's strategy dynamics. LILI does not explicitly optimize for stabilizing the opponent's strategy.

**Surprise Minimizing RL (SMiRL) [43].** SMiRL optimizes for a different notion of stability, where the reward is proportional to the likelihood of a state given the history of previously visited states.

**Stable.** We refer to Stable as the variant of SILI with $\beta = 1$, only optimizing for stability.

**SAC [45].** SAC does not model the opponent's strategy and is the base RL algorithm for SILI.

### 5.1 Simulated Environments

The environments and trajectories of stabilizing behavior are shown in the left column of Fig. 3. Across all figures, the ego agent's trajectory is drawn in blue, and grey corresponds to the opponent.

**Circle Point Mass.** In this environment, the ego agent is trying to get as close to the opponent as possible in a 2D plane, inspired by pursuit-evasion games [46]. The opponent moves between locations along the circumference of a circle. The ego agent never observes the true location (strategy) of the opponent. If the ego agent ends an interaction inside the circle, the opponent jumps counterclockwise to the next target location. If the ego agent ends an interaction outside the circle, the opponent stays at the red target location for the next interaction. We examine four variants: Circle (3 Goals), Circle (8 Goals), Circle (Continuous), Circle (Unequal). In Circle (Continuous), there are infinite possible opponent strategies, so we use continuous latent variables. In Circle (Unequal), the ego agent begins closer to two unstable opponent goals, but there is a stable goal farther away.

**Driving.** A fast ego agent is attempting to pass a slow opponent driver [10]. There are 3 lanes and a road hazard upcoming in the center lane, so both the ego agent and opponent need to merge to a new lane. If the ego agent merges to the left lane before the red line (giving the opponent enough reaction time), then the opponent merges to the right lane during the next interaction, understanding the convention of faster vehicles passing on the left. Otherwise, the opponent will aggressively try to cut off the ego agent by merging into the lane that the ego agent previously passed in.

**Sawyer-Reach.** The opponent can choose between three goals on a table with their intent hidden from the robot. The ego agent is the Sawyer robot that is trying to move their end effector as close as possible to the opponent's chosen goal in 3D space without ever directly observing the opponent's strategy [47]. If the end-effector ends the interaction above a fixed plane on the z-axis, the opponent's strategy stays fixed. Otherwise, the opponent's strategy changes. Semantically, we consider a robot server trying to place food on a dish for a human and needing to move its arm away from the dish by the end of the interaction to avoid intimidating the human retrieving the food.

**Detour Speaker-Listener.** In this environment, the agents must communicate to reach a goal, which has been a popular setting to explore emergent communication [18]. The speaker (opponent) does not move and observes the true goal of the listener (ego agent). The ego agent cannot speak, but must navigate to the correct goal. The speaker utters a message to refer to the goal, which the ego agent then observes. If the ego agent goes near the speaker (within some radius), the speaker follows the same communication strategy during the next interaction. Otherwise, the speaker chooses a random new strategy, or mapping of goal landmarks to communications. Critically, in this environment, the opponent's strategy is the communication strategy, not the true goal location.

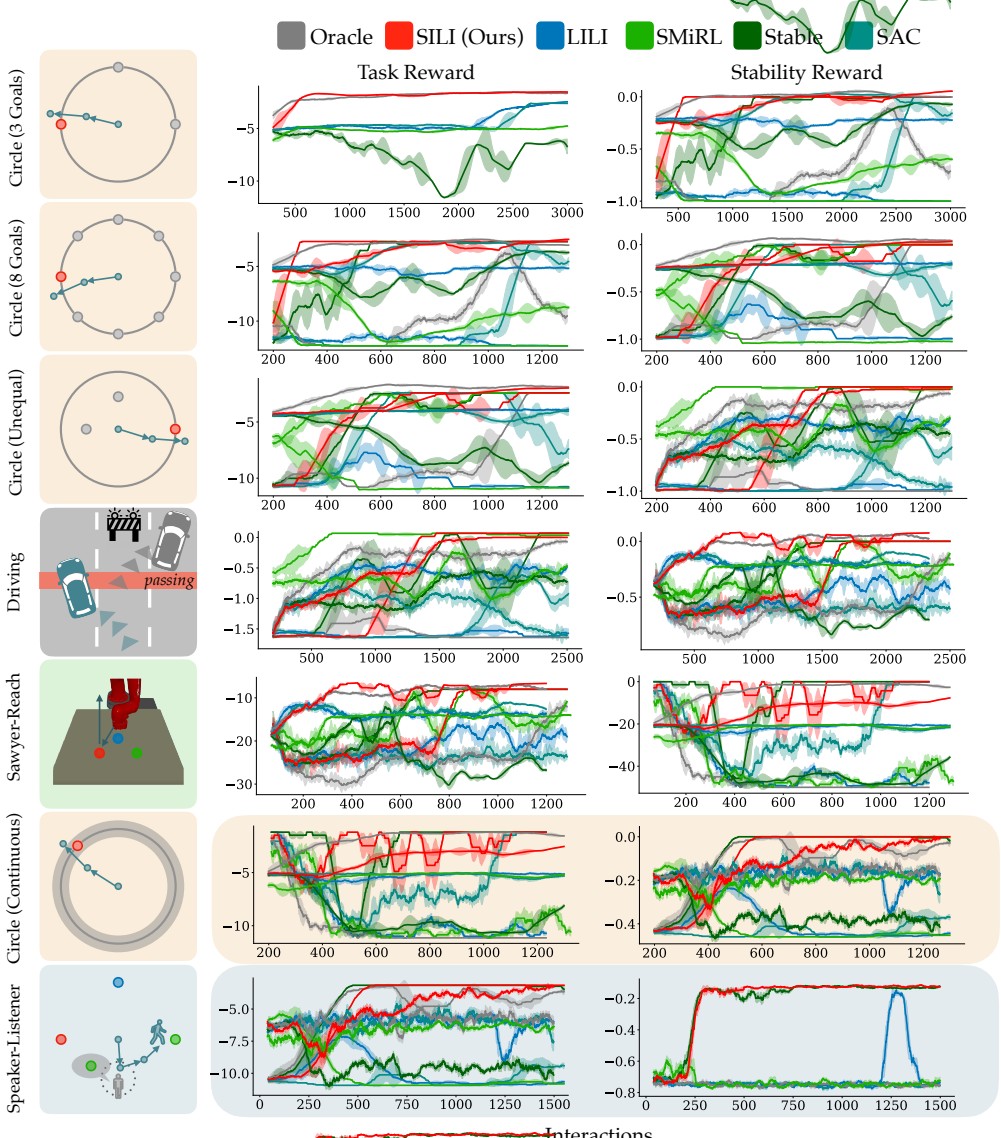

Figure 3: Task and stability rewards across all environments. SILI achieves comparable performance to Oracle. Compared to the other baselines, SILI significantly outperforms them in task reward (left plots) by learning to stabilize the opponent's strategy (right plots).

## 5.2 Results

Across all environments, without observations of the opponent's strategy, SILI achieves a comparable task reward to Oracle, as seen in Fig. 3. Further analysis is available in the Appendix and at https://sites.google.com/view/stable-marl/ with qualitative analysis of trajectories.

**Continuous Latent Variables.** We perform an experiment with continuous latent variables in the Circle (Continuous) environment (row 6, Fig. 3). Both SILI and Oracle learn to stabilize in this environment, although Oracle incurs various dips in stability reward at little to no cost to the task reward because Oracle knows the true opponent's strategy. While these results are promising, we observe in other environments that continuous latent variables and a smooth distance metric over the latent space lead to more noisy rewards, which hinder the ego agent's ability to learn effectively.

**Discrete Latent Variables.** We further show our results in a variety of environments with discrete latent variables. In Circle (3 Goals), SILI achieves the highest reward along with Oracle, but Oracle does not require stabilizing the opponent's strategy (row 1, Fig. 3). In Circle (8 Goals), we demonstrate that with more opponent strategies, SILI still learns to stabilize and outperforms the baselines (row 2, Fig. 3), where we set the latent dimension hyperparameter to be an overestimate of the true number of opponent strategies. In Circle (Unequal), SILI avoids trying to optimize for nearby, unstable opponent strategies, whereas the other baselines do not learn to stabilize since they greedily try to track the nearby, unstable opponent strategies (row 3, Fig. 3). In Driving, the task reward is based on the number of collisions with the opponent, and our method is the only one that learns to safely avoid all collisions (row 4, Fig. 3). In Sawyer-Reach, SILI and Oracle achieve comparable performance, but again, Oracle does not need to learn to stabilize (row 5, Fig. 3).

**Partial Observations of Changing Strategy.** To test our method in a setting without representation learning, the ego agent observes whether the opponent's strategy has changed between timesteps. We perform this experiment in the Detour Speaker-Listener environment, where the assumption that the opponent's strategy is kept fixed throughout an interaction is broken (row 7, Fig. 3).

SILI learns to stabilize and significantly outperforms all other baselines in task reward, including Oracle. In this environment, although Oracle can observe the true opponent strategy, the Oracle still needs to learn the association of the opponent's strategy and opponent's communication with the true goal. Further, the communication and strategy can change frequently between timesteps while the true goal stays constant, making learning difficult. The baseline Stable learns to stabilize but does not optimize for the task reward at all. The rest of the baselines learn to move towards the centroid of the goal locations since they cannot accurately decipher the opponent's strategy.

Notice that around interaction 250, SILI learns to stabilize, which requires moving in a detour away from the true goals, but SILI maintains the stability and learns to maximize task reward well. On the contrary, LILI temporarily learns to stabilize at around interaction 1250, but incurs a corresponding decrease in task reward, which discourages LILI from maintaining the stable opponent strategy.

## 6 Discussion

**Summary.** We propose a framework for learning in multi-agent environments that learns the dynamics of the opponent's strategies in a latent space. We then leverage this latent dynamics model to design an unsupervised stability reward, which can be augmented with the task reward to explicitly encourage the ego agent to learn how to stabilize and then more easily maximize the task reward.

**Limitations and Future Work.** In this work, we define a notion of pairwise stability that considers consecutive changes in the strategy. However, it may be interesting to consider smoother, longer horizon stability metrics, such as average change over a window of time. We compare our method to SMiRL, which optimizes for a probabilistic notion of state stability relative to states in an experience replay buffer, but show that SMiRL often does not learn to stabilize in a consistent manner. We plan to compare these different metrics of stability and their effects in future work. In addition, our experiments were mainly done in simulation. In the future, we plan to consider robot-robot and human-robot interactions, where we are required to learn much more complex dynamics that capture the non-stationarity of humans who adapt their strategies over repeated interactions. Stabilizing a human's strategy can greatly improve the efficiency in which a robot learns the best way to perform a task, but the exact method in which to stabilize human behavior should be further explored.

**Acknowledgments**

We would like to acknowledge NSF grants #1941722, #2006388, and #2125511, Office of Naval Research, Air Force Office of Scientific Research, and DARPA for their support.

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
