# OpenReview forum: "Influencing Towards Stable Multi-Agent Interactions"
_robot-learning.org/CoRL/2021/Conference — CoRL2021 Oral_

### Official Review · Reviewer_mhit · 2021-07-17

**Originality:** Very Good
**Technical Quality:** Good
**Clarity Of Presentation:** Very Good
**Impact:** 3

**Recommendation:**

Weak Accept: I recommend accepting the paper, but will not argue for my recommendation if the majority of other reviewers have a different opinion.

**Summary:**

In a two-player game, an ego-agent's reward can be augmented to encourage it to take actions that will stabilize the partner's strategy. The stability of the partner's strategy is defined by the change in a latent parameter that the ego-agent learns via unsupervised learning from the partner's previous trajectories.

**Issues:**

The authors should add a discussion of the limitation of the Hidden Parameter MDP formalism when applied to multi-agent tasks.

The authors should provide more detail about the dynamics of the opponent strategy for the given tasks.

The authors should consider avoiding the use of the word opponent, since most of the tasks presented are cooperative.

**Reviewer Expertise:**

Very good: Comprehensive knowledge of the area

**Strengths And Weaknesses:**

Stengths:
The idea of stability as presented in the paper is very clear. The implementation of the stability reward is also very intuitive and straightforward.
The authors provide an extensive review of related work in multi-agent learning. The comparison of stability and surprise is especially interesting.
The authors provide experimental results on several tasks that demonstrates that the SILI approach exceeds state of the performance of surprise-based algorithms.

Weaknesses & Questions:
The authors formalize a two-player game as a Hidden Parameter MDP, and leverage the assumptions of a static opponent strategy transition function. The application of this work to two-player games is poorly motivated.
I'm worried that this approach would have limited applicability in real human-robot interaction, and especially for competitive or adversarial tasks, in which an opponent may adapt ttheir policy or strategy to disguise their actions and win.
More discussion is needed about types of tasks in which it is impossible to stabilize the opponent's strategy.

Given strategy transition functions that are fixed, we can instead look at trajectories of N>>1 episodes in sequence. This new task may be considered an MDPs since the opponent's $T^z$ is stationary. One valuable baseline would be to apply SAC or standard RL algorithms to episodes that are composed of 10-100 interactions.

The authors may want to avoid the use of the term "opponent" since no adversarial tasks are used. Even in the Circle Point Mass task, the "opponent" has a stationary strategy and has no ability to adapt it to avoid capture.

For the Sawyer-Reach task, is the opponent's strategy the choice of goal?

The Detour Speaker-Listener task is the most interesting because the speaker's policy changes randomly if the ego-agent fails. What I'm surprised by is the inability of the Oracle to match the performance of SILI. This be due to the representation of the communication strategy provided to the agent.

Were there any challenges with tuning the relative magnitudes of the rewards, especially for not purely cooperative tasks?
Did the authors face any problems of convergence to sub-optimal policies? Seems like the stability reward could discourage exploration.

**Summary Of Recommendation:**

The authors apply the Hidden Parameter MDP formalism to two-player games, which is an extremely restrictive assumption as compared to the space of all two-player games. But given this problem formulation, the proposed algorithm is compelling and may be useful for partially observable control tasks.

---

> ### Author Response · Authors · 2021-08-26
> **Response to Reviewer mhit**
>
> We thank the reviewer for their constructive comments and detailed feedback. We are excited that the reviewer finds our idea of stability very clear, intuitive, and straightforward.
>
> **Competitive or adversarial tasks**
>
> Yes, we do agree that the framework of stabilizing the other agent’s strategy is more suitable for cooperative settings. That being said, stabilizing may still be applicable in competitive settings if the other agent is a programmed agent that has predictable strategy dynamics.
>
> **We can instead look at trajectories of N>>1 episodes in sequence. This new task may be considered an MDP since the opponent's Tz is stationary.**
>
> This is an interesting perspective. Indeed we can think about the MDP over strategies, and consider an episode’s worth of interaction as taking one step in that MDP. One worry is that exploring this MDP in full will not scale, which is why we propose the stabilizing framework to limit transitions in this MDP.
>
> **Avoid the use of the term "opponent"**
>
> Thank you for your suggestion. We can use the term “other agent” throughout the paper, if that is more clear. We have a footnote stating that we refer to the other agent interchangeably as the opponent, regardless of whether they are competitive or cooperative, following the convention of a survey on dealing with non-stationarity in multi-agent environments. We would be happy to revise this if that would make the paper more clear.
>
> **For the Sawyer-Reach task, is the opponent's strategy the choice of goal?**
>
> Yes, for the Sawyer-Reach task, the other agent's strategy is the choice of goal.
>
> **Inability of the Oracle to match the performance of SILI in Speaker-Listener**
>
> Although the Oracle is provided with the other agent's true strategy, the Oracle still must learn how to associate the communication and the other agent's strategy with the true goal, which becomes more difficult as the number of goals increases. With the other agent’s communication strategy stabilized, SILI can quickly learn how to interpret the other agent’s communication, and therefore learn which goal to move towards. This was initially surprising to us as well, but it really highlights the effectiveness of stabilizing strategies.
>
> **Were there any challenges with tuning the relative magnitudes of the rewards, especially for not purely cooperative tasks?**
>
> In our experiments, we found that a stability weight of \beta = 0.5 worked well for all of our experiments. In general, however, \beta should indeed be set based on the relative magnitudes of the rewards. We have added an ablation study in the Appendix D that discusses the impact of beta on our algorithm. In practice, one can anneal beta, starting from a high prioritization of stabilizing and then decaying to focus more on the task reward once the other agent’s strategy has stabilized.
>
> **Limitations of HiP-MDP formulation compared to POMDP**
>
> In our work, we are generally interested in the setting where our other agent’s strategy is unobserved directly by the ego agent. Thus, we can formalize our approach in a POMDP setting, where the observation augmented with the unobserved opponent strategy can be defined as the state. In our setting, we have a single hidden parameter (the other agent’s strategy), which stays constant throughout an interaction and updates between interactions, so this fits the HiP-MDP framework nicely. There likely exist other variants of the POMDP formulation that we could cast our problem into if desired. Specifically, if we wanted to experiment with model-based approaches, we could compare to algorithms for POMDPs, but we focus our work on model-free approaches, without knowledge of the underlying transitions and reward functions.

---

### Official Review · Reviewer_naZQ · 2021-07-20

**Originality:** Good
**Technical Quality:** Good
**Clarity Of Presentation:** Good
**Impact:** 2

**Recommendation:**

Weak Accept: I recommend accepting the paper, but will not argue for my recommendation if the majority of other reviewers have a different opinion.

**Summary:**

This paper proposes a novel reinforcement learning framework for multi-robot coordination. In this framework, the authors use a modified reward function that, in addition to maximizing long-term discounted reward, works to stabilize the other agents' policies to facilitate learning in the ego agent. They show through detailed experiments that this augmented reward function improves the agent's ability to coordinate and maximize the task reward.


**Issues:**

 The single figure used to display the results of comparing the proposed method across various environments is not descriptive enough to give a complete understanding of the framework's performance. More information on the results of these experiments would be helpful.

Minor comments
1) Real-world on page one should be hyphenated
2) Us and our teammate should be our teammate and us on page 2

**Reviewer Expertise:**

Good: General knowledge of the area

**Strengths And Weaknesses:**

The paper is well written and tackles a  relevant problem. The authors give a thorough description of their approach as well as experiments validating their claims. In addition, the authors provide clear discussion sections which summarize the results and discuss limitations.

However, the single figure used to display the results of comparing the proposed method across various environments is not descriptive enough to give a complete understanding of the framework's performance. More information on the results of these experiments would be helpful.

**Summary Of Recommendation:**

The paper is well written and tackles a well-studied but
still relevant problem. The authors provide a good
literature review and clearly explain their architecture,
presenting some experiments to support their claim.  I believe it is suitable for CoRL.

---

> ### Author Response · Authors · 2021-08-26
> **Response to Reviewer naZQ**
>
> We thank the reviewer for their helpful feedback and suggestions. We are excited that the reviewer thinks our paper tackles a relevant problem and that our experiments validate our claims. Here we address some of the concerns raised by the reviewer:
>
> **Qualitative Analysis Beyond Fig. 3**
>
> In regards to the single figure not being descriptive enough to give a complete understanding of the framework’s performance, we are not sure if you have had a chance to see some of our results including videos of rollouts of trajectories on the paper’s website: https://sites.google.com/view/stable-marl/? We provide qualitative analysis of our robot’s learned stabilizing behaviors compared to baselines. We have also added new results on the effect of the stability weight in Appendix D, as suggested by another reviewer.
>
> **Minor comments**
>
> 1. Real-world on page one should be hyphenated
> 2.Us and our teammate should be our teammate and us on page 2
>
> We have fixed the minor comments in the updated paper. Thank you for pointing those out!

---

> > ### Comment · Reviewer_naZQ · 2021-08-31
> > **Response to author's review**
> >
> > Thank you for the clarifications and congratulations on the nice work

---

### Official Review · Reviewer_CoBB · 2021-07-22

**Originality:** Good
**Technical Quality:** Good
**Clarity Of Presentation:** Very Good
**Impact:** 3

**Recommendation:**

Weak Accept: I recommend accepting the paper, but will not argue for my recommendation if the majority of other reviewers have a different opinion.

**Summary:**

This paper proposes to encourage agents to stabilize other agents thereby rendering the environment stationary and easier to learn. The underlying approach is to use an added reward function that encourages the agent to stabilize the other agent's strategy.
Experiments in various domains (e.g., driving, sawyer-reach, speaker-listener) suggest that the proposed SILI works well.


**Issues:**

The major issue for me was about how the latent space $\mathcal{Z}$ is learnt.

**Reviewer Expertise:**

Very good: Comprehensive knowledge of the area

**Strengths And Weaknesses:**

On the whole, the paper proposes a simple yet effective idea for stabilizing interactions between agents. To my knowledge, SILI is novel and the results are relevant for the robotics community, especially those in multi-agent learning and human-robot interaction. The paper is well-written on the whole (up to the points below). Extensive experiments in various domains validate the main claims and shows SILI works well against the baselines and unmodified SAC.

My only major concern is regarding the learnt latent strategy space $\mathcal{Z}$. The entire approach hinges upon a well-structured latent space, but the paper is vague about how exactly $\mathcal{Z}$ is learnt. Specifically, the authors state the approach is unsupervised but how are the data/trajectories/strategies sampled? It seems crucial that $\mathcal{Z}$ is sufficiently "smooth" and that nearby latent representations correspond to similar strategies; this can be challenging since only noisy trajectories are observed?

In more complex scenarios, it may be very difficult to learn a stabilizing strategy, especially with humans or sophisticated artificial agents. Again, this is contingent upon a good latent space and the inherent complexity of the other agent. While I think this paper is a step in the right direction, it's difficult to see if it will scale up.

As a minor comment, what stabilizing strategies were learnt in the various domains? I think qualitative descriptions would really help to increase the impact of the work. In particular, how do they differ to the non-stabilizing strategies? I understand that space is an issue but Fig 4 in the appendix should be in the main paper.



**Summary Of Recommendation:**

I lean towards acceptance. This work nicely executes a first-step towards more effective and stable interactions. I hope the authors can address my question above about the learnt latent strategy space.

### Post Response
Thanks for the response about the latent strategy space and for adding the qualitative results. Overall, I remain in favor of accepting this work.

---

> ### Author Response · Authors · 2021-08-26
> **Response to Reviewer CoBB**
>
> We thank the reviewer for their constructive feedback and detailed comments. We are excited that the reviewer thinks our work is a simple yet effective idea for stabilizing interactions between agents and relevant to the robotics community. Here we address the reviewer’s concerns:
>
> **Learning latent strategy space Z**.
>
> We completely agree that the approach is highly dependent on learning a well-structured latent space in order for the stability reward to be useful for influencing stable behaviors. Our framework allows our RL agent to learn the latent space by using its trajectories sampled during training. We use Soft Actor-Critic (SAC) as our base RL algorithm, so there is high entropy during the initial phases of exploration. With sufficient exploration, the RL agent will ideally collect a diverse set of trajectories and (unobserved) opponent strategies in its experience replay buffer, which can facilitate learning latent representations of the opponent’s strategy.
>
> We perform experiments in two settings with learned latent strategies. In the first setting, we allow the latent strategy space to be unconstrained and continuous, where we provide results for Circle Point Mass (Continuous). In this setting, the stability reward is defined in a continuous space, so it is important that small distances between latent strategies correspond to small distances between the true strategies. We found that precisely modeling the continuous latent strategy space could be difficult and lead to sub-optimal policies due to noisy representations, and thus noisy stability rewards. To address the difficulty of modeling a continuous latent strategy space, we examine a second setting with a discretized latent strategy space (through use of the Gumbel-Softmax method) where the RL agent only needs to be able to predict when an opponent’s strategy has changed, rather than precisely measure the amount of change. We found in our experiments that the discretized latent space provides much more consistent training, as it is easier to model discrete changes of the opponent’s strategy compared to measuring exact distances between consecutive strategies. Finally, we perform experiments in a setting where there is no learned latent strategy. Instead, we provide partial observations of when the opponent’s strategy has changed. In this case, we are demonstrating that our framework can benefit from improvements in representation learning, and that there is still merit in adding an auxiliary stability reward.
>
>
> **What stabilizing strategies were learnt in the various domains?**
>
> We provide qualitative results (rollouts of learned policies) which describe stabilizing and non-stabilizing behaviors at our website: https://sites.google.com/view/stable-marl/.

---

> > ### Comment · Reviewer_CoBB · 2021-09-01
> > **Thanks!**
> >
> > Thanks for the response about the latent strategy space and for adding the qualitative results. Overall, I remain in favor of accepting this work.

---

### Official Review · Reviewer_whdT · 2021-07-23

**Originality:** Good
**Technical Quality:** Good
**Clarity Of Presentation:** Very Good
**Impact:** 3

**Recommendation:**

Weak Accept: I recommend accepting the paper, but will not argue for my recommendation if the majority of other reviewers have a different opinion.

**Summary:**

This paper tackles the problem that other agents adapt/change their strategies/behaviors when interacting, resulting in a non-stationary policy. The authors propose to learn a policy that influences the other agents to not change their policy (i.e., to influence the other agents' strategy to stabilize) by penalizing the learning agent with a penalty function of the other agent's distance between the previous and current latent strategies. Hence, the main contributions of this work are:
1 - A stability reward function influencing the ego-agent to learn a policy minimizing the other agent's changes in their policy/behavior
2 - A strategy allowing to learn the stability reward function in an unsupervised manner

Moreover, this work presents simulation results demonstrating the claimed contributions.

**Issues:**

Clarity issues:
1 - How is the opponent strategy defined? Is z^j a sequence of actions over the rollout j?
2 - In line 145, in the definition of fixed-point strategy, how is \epsilon defined? What are the conditions on \epsilon for the definition to hold?

Notation problems:
1 - Line 157, the V function is not defined. What is V?
2 - Line 145 and182, N is not defined.

The related works section is missing an important reference and it is important to highlight how this work differs from the following references:
1 - Yang, Jiacheng et al. “Learning to Incentivize Other Learning Agents.” ArXiv abs/2006.06051 (2020)

Possible improvements:
1 - Ablation study and discussion on the impact of beta in the learned policy's performance.
2 -  How is the action space defined for each experiment? In some experiments, it is not clear.
3 - Adding qualitative results in the paper or attach a video clearly showing the learned policy in action.



**Reviewer Expertise:**

Very good: Comprehensive knowledge of the area

**Strengths And Weaknesses:**

In general, this paper is very well-written and tackles an important problem to the research community. The contributions and proposed work's motivation are clearly written, the problem statement is well-defined, and the presented results support the work claims.

The main strengths of this paper are:
1 - The idea of rewarding the learning agent to stabilize the other's strategies when interacting.
2 - The proposed method allowing to learn the stability reward in an unsupervised manner.

The main weaknesses of this work are:
1 - No video displaying the qualitative results of the proposed method. Adding visual qualitative results can improve the significance of this work and can be important to make a good qualitative evaluation of the method. The results presented in the appendix do not a whole episode.
2 - The code used for the experiments is not provided. Unfortunately, it is more and more important to release the code used to generate the presented results and allow others to reproduce presented results to avoid people getting articles with false claims published. Open-sourcing the code used for the proposed method will strengthen the contribution of this work and make future works cite it as a reliable reference.
3 - The experiments are still very simple and far from a real robotic application scenario.

**Summary Of Recommendation:**

To summarize, I think this paper tackles an important problem and the proposed method has some novelty. Moreover, the paper is very well-written. However, adding more complex experiments and qualitative results showing the learned policy in action would definitely increase this work contribution. Still, as a conference paper, I think it is in a good state to be accepted.

---

> ### Author Response · Authors · 2021-08-26
> **Response to Reviewer whdT**
>
> Thank you so much for your thoughtful feedback. We are excited that you found our work an important problem to the research community and that our presented results support our work’s claims. We would like to address the reviewer’s concerns here:
>
> **Qualitative results**
>
> We have provided GIFs of simulation results and further qualitative analysis at this website, which we have linked in the paper: https://sites.google.com/view/stable-marl/.
>
> **Source code**
>
> We have also provided a link to download our anonymized code from the project website. We wholeheartedly agree that reproducibility is a critical facet of research.
>
> **How is opponent strategy defined?**
>
> The opponent’s strategy for an interaction j is denoted as z^j, which is the hidden parameter in the hidden-parameter MDP. z^j is an unobserved variable that is constant throughout interaction j. For example, in our Circle (3 Goals) environment, the opponent’s strategy is the goal location, which is constant throughout one interaction, but changes between interactions.
>
> **How is \epsilon defined?**
>
> In line 145, we are reiterating our definition of pairwise \epsilon-stable, so the \epsilon is defined depending on how strictly one would like to enforce stability. \epsilon determines the strictness of the pairwise \epsilon-stable definition.
>
> **Notation problems: 1 - Line 157, what is V? 2 - In Line 145 and 182, what is N?**
>
> V is the value function of the HiP-MDP. N is the total number of interactions. We have added clarifications for both of these in the main body of the paper.
>
> **Missing related work: Yang, Jiacheng et al. “Learning to Incentivize Other Learning Agents.” ArXiv abs/2006.06051 (2020)**
>
> This is an interesting line of work and we have added this reference to our paper. We have added the suggested citation as well as similar citations along the lines of peer-rewarding mechanisms: 1 - “Gifting in Multi-Agent Reinforcement Learning.” https://dl.acm.org/doi/abs/10.5555/3398761.3398855 and 2 - “Emergent Prosociality in Multi-Agent Games Through Gifting.” https://arxiv.org/abs/2105.06593. Our work shares similar motivations in trying to encourage cooperative, prosocial behaviors in multi-agent environments. However, our work differs in that we do not assume that an external mechanism exists that allows agents to share their rewards with each other. Instead, our SILI agent must learn a model of the opponent’s strategy’s dynamics conditioned on their own behavior.
>
> **Impact of beta**
>
> We have added an ablation study in Appendix D that discusses the impact of beta on our algorithm. In our environments, we found that a good choice of beta roughly balanced the average task and stability rewards. In practice, one can anneal beta, starting from a high prioritization of stabilizing and then decaying to focus more on the task reward once the opponent’s strategy has stabilized.
>
> **Action space descriptions**
>
> We have provided more details of each environment’s action space in Appendix B: Further Details of Simulated Environments.

---

> > ### Comment · Reviewer_whdT · 2021-08-30
> > **Response to author's review**
> >
> > Great! Thanks for the clarifications and congratulations on the nice work!

---

### Meta-Review · Area_Chair_bzTW · 2021-08-14

**Recommendation:** Accept (Oral)
**Confidence:** 4

**Metareview:**

Overall, this work tackles an important problem in multi-agent domain. The idea proposed in this paper is also interesting together with the promising results in experiments. Reviewers raised consistent comments that this paper is well-written and organized, and the novelty also makes sense. However, reviewers also care about a lot, such as releasing the video results and source codes. The experimental setting is a little bit simplified, which sounds like still far away from the real world scenarios. Authors should also explain more details about the learning procedure.

During the rebuttal period, authors have replied most of questions from reviewers. Most of reviewers think this is a nice work and recommend the acceptance.

---

> ### Author Response · Authors · 2021-08-18
> **Clarifying question about qualitative results on website**
>
> Thank you so much for taking the time to review our work. We are working on addressing the reviewers’ questions in detail to clarify any questions. Our website (https://sites.google.com/view/stable-marl/) contains qualitative analysis and GIFs of learned robot behaviors in our various experiment settings. We will more clearly provide the link to our website in the abstract of our updated paper. We will also provide a link to anonymized source code on our website. Given the comments on providing qualitative videos, we suspect that we did not adequately point to these qualitative results on our website. We wanted to check and ask if the meta-reviewer and reviewers are interested in seeing more qualitative analysis beyond what is provided on the website?

---

> ### Author Response · Authors · 2021-08-26
> **Meta-reviewer Response**
>
> We thank the meta-reviewer and all the reviewers for their constructive feedback and helpful suggestions. We appreciate that the reviewers agree that the work tackles an important problem in the multi-agent domain and provides a first step towards “more effective and stable interactions.” The reviewers mainly have questions about video results, source code, and clarifications of the experiment settings.
>
> **Video results**
>
> In our responses to the reviewers and in our updated paper, we have clarified that our website displays qualitative results (rollouts of learned policies) which describe learned stabilizing and non-stabilizing behaviors: https://sites.google.com/view/stable-marl/.
>
> **Source code**
>
> We have released a link to anonymized source code on our website as well.
>
> **Clarification of experiments**
>
> We discuss more about the learning procedure of the latent strategy space, as well as address minor comments suggested by the reviewers. We further provide an extra analysis into the effect of the stability weight in Appendix D. In our updated paper, we use red text to denote the revised text.

---

### Decision · Program_Chairs · 2021-09-13

**Decision:**

Accept (Oral)

**Comment:**

Overall, this work tackles an important problem in multi-agent domain. The idea proposed in this paper is also interesting together with the promising results in experiments. Reviewers raised consistent comments that this paper is well-written and organized, and the novelty also makes sense. However, reviewers also care about a lot, such as releasing the video results and source codes. The experimental setting is a little bit simplified, which sounds like still far away from the real world scenarios. Authors should also explain more details about the learning procedure.

During the rebuttal period, authors have replied most of questions from reviewers. Most of reviewers think this is a nice work and recommend the acceptance.